# Biopsychosocial and Nutritional Factors of Depression among Type 2 Diabetes Mellitus Patients: A Systematic Review

**DOI:** 10.3390/ijerph19084888

**Published:** 2022-04-17

**Authors:** Norizzati Amsah, Zaleha Md Isa, Norfazilah Ahmad

**Affiliations:** Department of Community Health, Faculty of Medicine, Universiti Kebangsaan Malaysia, Jalan Yaacob Latiff, Bandar Tun Razak, Kuala Lumpur 56000, Malaysia; norizzati1988@gmail.com (N.A.); norfazilah@ppukm.ukm.edu.my (N.A.)

**Keywords:** Type 2 Diabetes Mellitus, risk factors, determinants, depression, mental health

## Abstract

The rising prevalence of depression among Type 2 Diabetes Mellitus (T2DM) patients has triggered an alarming situation, and further actions need to be taken by health care professionals and policymakers to curb the issue. There is a lack of evidence review in terms of the biopsychosocial and nutritional factors that are related to depression among T2DM. Hence, this review aimed to identify available evidence on the biopsychosocial and nutritional factors associated with depression among T2DM patients based on the existing literature. Articles were systematically searched from four databases, namely PubMed, Scopus, Web of Science, and EBSCOHost. The approach for the identification of the final articles followed PRISMA guidelines. The selected full-text articles were published between 2017 and 2021 in the English language, and included studies focused on depression among T2DM patients. Using AXIS tools, the eligible articles were evaluated to assess the quality of studies. A total of 19 studies were included in the review, and information related to research questions and associated factors was extracted. Biological, psychological, social, and nutritional factors were shown to be linked with depression among T2DM patients. Future studies need to considered using the biopsychosocial model and incorporating nutritional factors to manage the issues of depression among T2DM patients.

## 1. Introduction

Type 2 Diabetes Mellitus (T2DM) is an emerging public health issue with a high prevalence worldwide. The prevalence of T2DM is increasing, with a significant increase in low- and middle-income countries, including Southeast Asian countries [1]. T2DM a health challenge and has affected 415 million patients globally [2]. Moreover, it is predicted that by 2040, the number of people with T2DM will increase to 642 million worldwide [3]. Recently, mental health among T2DM patients has become a topic of clinical discussion and policymaking [4].

Despite microvascular and macrovascular complications, T2DM was found to negatively affect the mental health status of patients, contributing to mental health outcomes such as depression [5]. Moreover, T2DM patients had a three-times higher risk of developing depression than those who did not have Diabetes Mellitus [6]. Individuals with T2DM and depression were found to have more severe long-term effects than those who did not have T2DM [5]. Depression was found to affect patients in terms of (i) quality of life, (ii) economic impact, and (iii) morbidity and mortality. In terms of quality of life, depression influences adherence to treatment and indirectly contributes to poor sugar control, as well as reduced productivity [7]. In addition, depression among T2DM patients was found to increase the risk of developing coronary heart disease up to 36.8% and the risk of heart-related deaths around 47.9% [2,7].

According to a biopsychosocial model developed by George Engel in 1977, the development of illness or disease occurs through a combination of biological factors, namely genetic and physiological [8]. For psychological factors, the development of illness or disease is linked to health beliefs, lifestyle, personality, and behaviour. In addition, social factors include family relationships, socioeconomic status and social support [8]. Thus, this model suggests that biological, psychological, and social factors are associated in the development of illness [9].

Based on the previous literature, nutritional factors play an important role in mental health. Poor dietary habits and nutritional factors may put T2DM patients at a higher risk for developing depressive symptoms [10]. Studies conducted in Iran have shown that depression was associated with low protein and high fat intake in an individual dietary pattern [11]. T2DM patients with low carbohydrate diet showed a significant association with mental health status [12]. Otherwise, there is still a lack of reviews discussing in detail biopsychosocial and nutritional factors in relation to depression among T2DM patients. Therefore, this review aimed to identify available evidence on biopsychosocial and nutritional factors associated with depression among T2DM patients.

## 2. Materials and Methods

This review was conducted using the Preferred Reporting Items for Systematic Reviews and Meta-Analyses (PRISMA) review protocol [13]. NAM and ZMI started the review by formulating appropriate research question. The process conducted for systematic searching consists of identification, screening, and eligibility processes. During the identification process, four primary databases, namely Scopus, Web of Science, PubMed, and EBSCOHost, were used. Most articles were open-access articles, which provided full original articles. Full original articles were selected to extract the relevant information and findings to answer the research question. In term of the quality of the selected articles, the articles were screened and evaluated using the Appraisal tool for Cross-Sectional Studies (AXIS) [14].

### 2.1. Formulation of the Research Question

In this review, the formulation of research questions was based on the PICo concept, which stands for the population, interest, and context [15]. This tool is used to guide authors in forming a suitable research question for the systematic review [15]. These components are the crucial elements in constructing the research question. Based on this concept, “population” refers to Type 2 Diabetes Mellitus patients, “interest” refers to depression, and “context” refers to biopsychosocial and nutritional factors or determinants. The PICo guided the formulation of the main research question in this review: “What are the biopsychosocial and nutritional factors associated with depression among T2DM patients?”

### 2.2. Systematic Searching Strategies

The systematic searching strategy in this review is based on PRISMA flow, which consists of the identification, screening, and eligibility stages (Figure 1).

### 2.3. Identification

During the identification stage, keywords were chosen using synonyms and variations that were found when searching for articles in databases. All authors (NAM, ZMI, and NA) screened the titles and abstracts independently according to their relevance based on the research question. The search string was use in the identification process, as shown in Table 1. The systematic literature search was conducted between 20 December and 27 December 2021, and involved four primary databases, namely Scopus, Web of Science, PubMed, and EBSCOHost, which resulted in the retrieval of 418 records. The records were retrieved from the databases and organised in an Excel sheet for screening. During the procedure, any duplicated articles and unrelated titles that deviated from the study question were deleted. Following that, the abstracts were reviewed before the unrelated articles were removed.

### 2.4. Screening Using Inclusion and Exclusion Criteria

The title and abstract of each article were examined by all authors (NAM, ZMI, and NA) for relevance based on the inclusion and exclusion criteria for this review. The inclusion criteria for article selection were: (1) published between 2017 and 2021, (2) full original article, (3) written in English, and (4) study focused on identifying factors associated with depression among T2DM patients. Systematic reviews, conference proceedings, book chapters, editorial letters, and reports were excluded. The screening process excluded 319 articles, and the retrieval of full-text eligibility proceeded for the remaining 35 articles.

### 2.5. Eligibility

A total of 30 full text articles were successfully retrieved for eligibility. NAM, ZMI, and NA reviewed all full-text articles and removed non-related articles. The remaining 19 articles were reviewed with the quality appraisal process.

### 2.6. Quality Assessment

The Appraisal tool for Cross-Sectional Studies (AXIS tool) is designed for non-experimental research and includes 20 components measuring each aspect of the study’s quality [14]. This tool determines whether the findings of a study are credible and reliable, and should related to the aims, methods, and analysis of what is reported and not on the interpretation (e.g., discussion and conclusion) of the study. This tool does not provide a numerical scale for evaluating the quality of the study. The benefit of the AXIS tool is that it provides the opportunity to assess each aspect of study design to provide an overall assessment of the quality of the study. This tool offers more flexibility in incorporating quality of reporting and risk of bias when making assessment on the quality of a paper. Based on the assessment, a total of 19 articles were included. The total number of “yes” appraisals was recorded for each study, as the tool guide does not specify a standardized scoring measure or numerical scale. For this review, the mean total quality score was 16 (range: 15–17) out of 20, as presented in Table A1 and Table A2 in Appendix A. The authors were able to conclude that included (19 studies) had clear study objectives and used the appropriate study design and methods.

### 2.7. Data Abstraction and Analysis

The authors independently extracted information from the selected studies, including the author’s names, year, country, study objectives, study designs, sample size, depression instrument, factors associated with depression among T2DM patients, and limitations. Data abstracted from all studies were compiled in an appropriate matrix table (Table 2). The authors (NAM, ZMI, and NA) reviewed the matrix tables for both consistencies and inconsistencies to generate themes and findings for the review. Similar or related information were grouped together as one characteristic, and the technique was repeated to form reasonable findings for interpretation.

## 3. Results

After the screening and selection process, a total of 19 studies which fulfilled the inclusion criteria were included in this systematic review (Table 2). The selected articles comprised four articles from India, three articles from Saudi Arabia, two articles from Malaysia, and two articles from Ethiopia. The remaining articles were from China, Korea, Nepal, Kuwait, Singapore, Turkey, and Pakistan, respectively. Eighteen articles of these quantitative studies were cross-sectional studies, and one was a cohort study. The review found various measurements used for depression and its related factors among T2DM patients (Table 2). The extracted articles were published from 2017 to 2021.

### Factors Associated with Depression among T2DM Patients

All findings were summarized into biological, psychological, social, and nutritional factors in Table 3. The review showed that age, the female sex, a long duration of diabetes, high body mass index, high HbA1c levels, the presence of comorbidities, diabetic complications, a mild impairment of cognitive function, and sexual dysfunction were biological factors. Psychological factors included the fear of diabetic complications, anxiety, low satisfaction with treatment, and physical inactivity. For social factors, depression was associated with alcohol intake, poor social support, low educational status, smoking habits, being single, living arrangement, and low socioeconomic status. As for nutritional status, low albumin was with depression among T2DM patients.

## 4. Discussion

To our knowledge, this is the first review discussing the factors associated with depression among T2DM patients based on the biopsychosocial model. This review comprised 19 articles identified multiple factors associated with depression among T2DM patients. These factors were then further divided into four major categories, namely (1) biological factors, (2) psychological factors, (3) social factors, and (4) nutritional factors.

### 4.1. Biological Factors

Based on the biopsychosocial model, biological factors, including genetic and physiological factors, cause the occurrence of a disease [8]. Referring to a study conducted by Habtewold et al. [4] which used this model, the biological factors included age, gender, fasting blood sugar levels, body mass index, comorbidity disease, and diabetes complications [4]. In terms of age (less than 65 years old), the female sex was among the contributing factors in the incidence of depression among people with T2DM [34]. Based on previous studies, patients who were older and had various chronic diseases faced more stress [35]. In terms of gender, women had a higher risk and were more prone to depression and anxiety compared to men [20,36].

A study in the United States showed that various comorbidities increase the risk of depression among T2DM patients [37]. The presence of comorbidities, such as hypertension and coronary heart disease, is among the contributing factors [22,38]. In the long term, T2DM may cause various complications, such as diabetes nephropathy, neuropathy, and retinopathy. The presence of complications has caused the patients to face more stress related to their health condition, thus contributing to depression and anxiety [20,33]. Diabetic complications, such as diabetic retinopathy, diabetic nephropathy, and neuropathy, predispose patients to depression. Studies have shown that the positive association between diabetes complications and depressive symptoms is persistent and mediated by diabetes distress [39]. Diabetic complications occur due to the long duration of diabetes. In other aspects, a long period of illness is also a contributing factor of depression among T2DM patients [33].

Metabolic problems in T2DM may contribute to the incidence of depression. Persistent hyperglycaemia and severe hypoglycaemia among T2DM patients can alter function, neuronal transmission, or brain structure, hence predisposing T2DM patients to depressive disorder [40]. Obesity is linked to anxiety and depression [25]. Obesity has been found to negatively affect self-confidence, economic status, and social status, and further contributes to psychological problems [40]. One study showed an association between obesity and depression due to the presence of genes contributing to depression, such as genes that encode glucocorticoids, leptin, and dopamine receptors [41]. Depression was associated with an increased risk of macrovascular complications, mortality, suicide, and all-cause mortality among patients with incident T2DM [42].

In other aspects, sexual disorders among T2DM patients have a significant association with the incidence of depression. Sexual disorders are more common in diabetics than non-diabetics [43]. Sexual disorders were more frequent among older T2DM patients and those with a longer duration of diabetes [43]. In addition, sexual dysfunction can negatively impact marital relationships and treatment outcomes [44].

In addition, depression, anxiety, and apathy are prevalent neuropsychiatric symptoms in mild cognitive impairment, and they have been associated with cognitive and functional deterioration in daily tasks, as well as disease progression [45]. In individuals with diabetes, depression is associated with a higher risk of dementia and poor cognition [46].

### 4.2. Psychological Factors

Psychological factors involve health beliefs and lifestyle [9]. The psychological factors in the study conducted by Habtewold et al. [4] included physical activity and the fear of complications of the disease and treatment. The fear of diabetic complications and anxiety also play an important role in depression among T2DM patients. Patients with anxiety symptoms are more inclined toward depression [6]. In addition, patients not only feel burdened by the disease but also fear further Diabetes Mellitus complications [22]. The increased burden of treatment cost is also among the psychological factors that contribute to anxiety and depression among patients with T2DM [47].

Life satisfaction is a necessary component in achieving positive mental health and is a predictor of several life outcomes [48]. Low satisfaction with treatment was shown a predisposing factor for depression among T2DM patients [24]. Patients with T2DM having moderate satisfaction with their treatments had greater medication adherence and a better quality of life [49]. Physician–patient interaction satisfaction is an important outcome in health care delivery, and most important determinant for patient satisfaction and medication adherence [50].

Generally, physical inactivity is linked with a higher mortality rate, as well as coronary disease, stroke, T2DM, and poor mental health [51]. Physical activity plays an important role in preventing depression [52]. A previous study showed that a sedentary lifestyle is a contributing factor to depression [19]. According to World Health Organization guidelines, adults aged between 18 and 64 years old should perform at least 150 min of moderate-intensity physical activity throughout the week for at least 5 days each week and for at least 30 min per day [53]. Excessive high-intensity physical activity can lead to fatigue and increase the risk of depression. Moreover, moderate-intensity physical activity is ideal to improve quality of life and can reduce the prevalence of depression [25]. Promoting exercise in addition to drug treatment has been shown to offer significant advantages in managing the symptoms of depression [54]. Exercise has therapeutic effects on depression among all age groups, mainly those between 18 and 65 years old, as a monotherapy, adjuvant therapy, or combination therapy in which the advantages of exercise therapy are comparable to conventional depression treatments [55].

### 4.3. Social Factors

Based on the biopsychosocial model, social factors consist of socioeconomic status, education level, marital status, and social support [4] In terms of financial status, patients from low-income families have a higher risk of depression [7]. Low socioeconomic status, poor education, and divorce are among the factors involved in the incidence of depression [38]. For employment status, unemployed individuals are more prone to the risk of anxiety and depression [56]. This is expected, as not having work is itself a depressing factor. In addition, the disease itself imposes a financial burden [57]. The association between lower educational status and T2DM can be attributed to a lack of understanding of the disease’s severity, drug compliance, and dietary management, which makes individuals vulnerable to complications [57].

Unhealthy lifestyles, such as smoking and the consumption of alcohol, are highly interrelated with depression, whereby depression may elicit and exacerbate unhealthy lifestyles. Moreover, people with unhealthy lifestyles are more likely to become depressed over time [58]. In a longitudinal study, smoking on a daily and non-daily basis and heavy alcohol consumption were associated with depression [58].

Loneliness is one of the risk factors for T2DM [59]. Being single was associated with depression among T2DM patients [19]. Living arrangement also plays an important role. Living arrangements, such as living alone, may be related to the incidence of T2DM [60]. Patients who lived with their children were found to be significantly less depressed compared to those who lived alone [28]. It has been postulated that living with children has a protective effect from depression, as patients receive good support and attention from the children.

Social support is very important to maintain good physical and mental health. Overall, positive and high-quality social support is crucial to prevent any occurrence of depression and mental stress [61]. Previous studies have shown that poor social support is significantly associated with depressive symptoms. Individuals with poor social support from family, neighbours, and friends are six-times more susceptible to depressive symptoms than individuals with strong social support [6]. The coping mechanism is crucial in order to adapt to any stressful situation [62]. T2DM is associated with a high prevalence of depression, which is influenced by personality factors and coping styles [63]. Poor coping mechanisms predispose T2DM patients to a higher incidence of depression [63]. Negative coping methods, such as protest or isolation, were associated with a lower quality of life, whereas avoidance was associated with increased diabetes-related distress and depressive symptoms [62]. Those who used positive coping techniques, and demonstrated high levels of personal and treatment control had lower levels of depression [64]. Other reviews have suggested that T2DM patients may benefit from psychoeducational interventions, such as supportive psychotherapy, in managing the complications of diabetes [63].

### 4.4. Nutritional Factors

Nutritional factors and dietary habits can affect physical health and illnesses [65]. The percentage of macronutrients consumed in daily life may be associated with the occurrence of depression [65]. High fruit and vegetable consumption was associated with a decreased incidence of T2DM, hypertension, dyslipidaemia, osteoarthritis, and depression in both men and women [66]. Studies in the United States have shown an association between protein and fat intake and depression [67]. Carbohydrates are one of the sources of energy for the population in Asian countries [68]. Carbohydrates play an important role in controlling sugar levels, and contribute to the incidence of T2DM and complications of Diabetes Mellitus [12].

Overall, a high amount of calorie intake and increased sugar intake in food were found to correlate with anxiety and depression symptoms [69]. Patients who consume fewer carbohydrates have better sleep quality and are less likely to suffer from mental illnesses [12]. Studies conducted in Iran have shown that depression is associated with low protein and high fat intake in individual dietary patterns [11]. A study showed that total protein intake and protein intake from milk and milk products might reduce the risk of depressive symptoms, but protein intake from red meat, poultry, fish, grain products, and legumes did not have the same result [70].

In this review, lower albumin levels due to poor nutritional status contributed to depression among T2DM patients [29]. Albumin is a protein that supports a variety of functions in the human body. Many studies have found that lower serum albumin levels are related to depression in distinct groups of psychiatric and non-psychiatric patients [71]. As summarised, nutritional factors play a crucial role in terms of mental health, especially depression, among T2DM patients.

### 4.5. Strength and Limitations

This review provides an overview of the biopsychosocial and nutritional factors that are associated with depression among T2DM patients based on the existing literature. In this review, the research was limited to articles published in English only. The systematic literature only extracted articles from 2017 to 2021. Most of the studies were observational studies, including cross-sectional studies and one cohort study. Despite being conducted using PRISMA guidelines for systematic reviews and using four different databases to perform the systematic search, this study still has some limitations. The review included only open-access articles; thus, it may not represent all available literature linked to the topic. In addition, some studies may have focused on similar topics that were eliminated during the screening process due to the different keywords and titles used by the studies.

## 5. Conclusions

This systematic review identified multiple factors that contributed to depression among T2DM patients from multiple countries. The findings revealed that biopsychosocial and nutritional factors had a greater impact on depression among T2DM patients. All identified factors can facilitate health care professionals in taking the necessary preventive measures. Using the biopsychosocial model, a better and more holistic approach can be devised to manage the issues of depression among T2DM patients.

## Figures and Tables

**Figure 1 ijerph-19-04888-f001:**
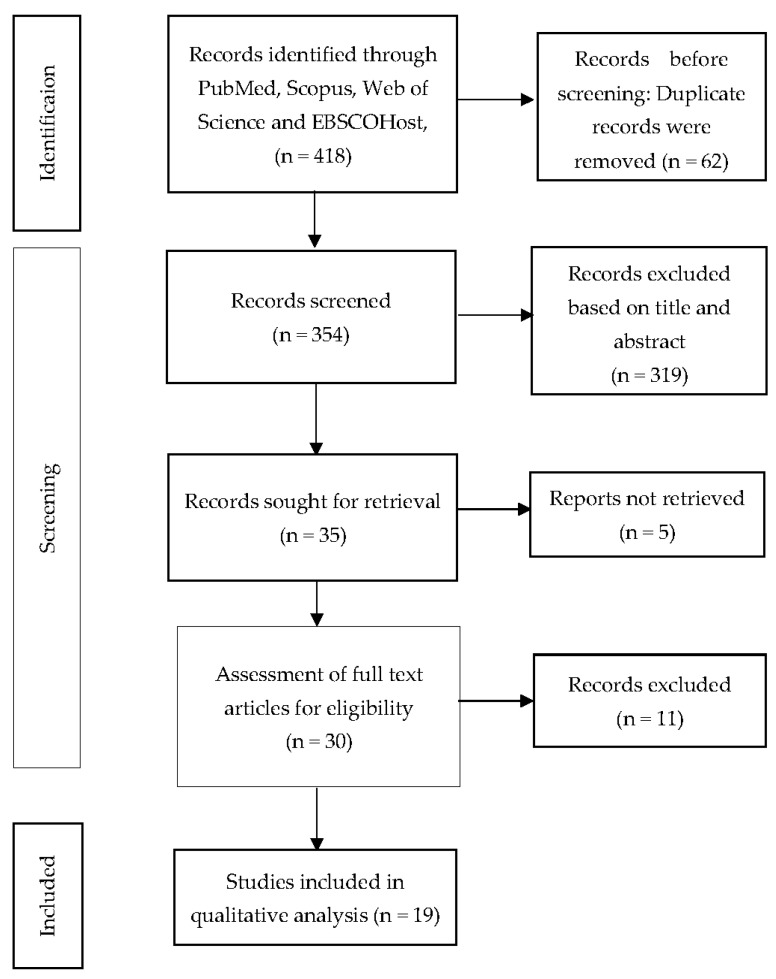
PRISMA flow diagram.

**Table 1 ijerph-19-04888-t001:** Keyword search in the identification process.

Database	Search String
Scopus	TITLE = (“mental health” OR “depression *” OR “depressive disorder *”) AND (“determinants” OR “predictors” OR “risk factors”) AND (“diabetes mellitus *” OR “hyperglycaemia” OR “glucose intolerance”)
Web of Science	TS = (“mental health” OR “depression *” OR “depressive disorder *”) AND (“determinants” OR “predictors” OR “risk factors”) AND (“diabetes mellitus *” OR “hyperglycaemia” OR “glucose intolerance”)
PubMed	(“mental health” OR “depression *” OR “depressive disorder *”) AND (“determinants” OR “predictors” OR “risk factors”) AND (“diabetes mellitus *” OR “hyperglycaemia” OR “glucose intolerance”)
EBSCOHost	(“mental health” OR “depression *” OR “depressive disorder *”) AND (“determinants” OR “predictors” OR “risk factors”) AND (“diabetes mellitus *” OR “hyperglycaemia” OR “glucose intolerance”)

The symbol * was used as truncation and wildcard function to increase the variability of selected keywords.

**Table 2 ijerph-19-04888-t002:** The measurement of depression and its related factors among T2DM.

No.	Authors, Country	Study Design	Objectives	Sample Size	Depression Measurement	Associated Factors	Associated Factors (Group Classification)	Limitation
1.	Pal et al. [16], India	Cross-sectional study	To assess the prevalence, severity, and determinants of depression among patients with T2DM	290 patients with T2DM, age > 18 years.	Diagnostic and Statistical Manual of Mental Disorders, Fifth Edition (DSM-5) criteria. Hamilton Depression Rating Scale (HAM-D)	Female and diabetic retinopathy	Biological	(1) Hospital-based study with convenient sampling design.(2) Selection bias, as sick patients and those with complications are more likely to visit tertiary care hospitals. (3) It was a cross-sectional study.
2.	Majumdar et al. [17], India	Cross-sectional study	To assess the predictors of depression as well as its prevalence in T2DM patients, the authors conducted a cross sectional study entitled ‘‘DEPression in DIABetes” (DEPDIAB).	1371 T2DM patients from Eastern India	Patient Health Questionnaire-9 (PHQ–9) and Beck depression scales	Younger age (18–40), female, low socioeconomic status, poor compliance, hypoglycaemia, and difficulty in managing day-to-day activities	Biological, Social	(1) It was conducted in the tertiary care centres and not in the community set-up, which may limit its generalizability. (2) The assessment of depression was performed via a questionnaire-based screening tool rather than by a psychiatrist.
3.	Kant et al. [18], India	Cross-sectional study	This study explores the predictors for depression in patients with T2DM	250 subjects at the diabetic clinic and psychiatry outpatient department of a tertiary care teaching hospital	The Center for Epidemiological Studies (CES-D)	Age, gender, locality, BMI, and FBS among diabetic patients	Biological	It was a cross-sectional and hospital-based study.
4.	Al-Ozairi et al. [19], Kuwait	Cross-sectional study	This study aimed to describe the prevalence of and risk factors for depression and diabetes distress in people with T2DM and whether depression and distress are independently associated with worse biomedical outcomes.	465 patients	PAID, Patient Health Questionnaire-9 (PHQ-9)	Sociodemographic (age, gender, and marital status) and body mass index, HbA1c	Social, Biological	Cross-sectional study
5.	Gebre et al. [20], Ethiopia	Cross-sectional study	The aim was to assess severity of depression and its determinants in diabetes outpatients at Hawassa University Comprehensive Specialized Referral Hospital, southern Ethiopia.	688 patients	Patient Health Questionnaire-9 (PHQ-9)	Consumption of alcohol, failure to practice recommended physical activity, not practicing a recommended dietary regimen. Loss of very close relative/spouse, poor social support.	Biological, Social, Psychological, Nutritional	(1) Recall bias for the questions like the PHQ scale, subjective and social desirability bias for questions about alcohol intake, physical activity, dietary regimen, and medication.
6.	Gupta et al. [21], India	Cross-sectional study	To determine the prevalence and predictors of depression in patients of DM among various sociodemographic, clinical, and quality-of-life variables	300 patients from outpatient department and inpatient department of a secondary care centre of Northern India	Hindi version of Patient Health Questionnaire-9 (PHQ-9)	Poor education	Social	(1) Small sample size, (2) an assessment of depression through a self-reported questionnaire. Hindi translation of the QOLID has not been validated.
7.	Abate and Gedamu [6], Ethiopia	Cross-sectional study	To identify psychosocial and clinical factors associated to develop depression symptoms in diabetes patients	416 patients	Patient Health Questionnaire-9 (PHQ-9)	Fear of diabetic complications, social support, being female, and sexual dysfunction	Psychological, biological	Using self-reported question
8.	Woon et al. [22], Malaysia	Cross-sectional study	To determine the prevalence of depression and anxiety, and their associated factors in the Malaysian diabetic population	300 diabetic patients	The Beck Depression Inventory (BDI)	Anxiety, which increased the occurrence of depression by almost 20-fold	Psychological	(1) A single tertiary healthcare referral centre. (2) The depressive and anxiety symptoms were measured by self-reported tools rather than diagnostic interviews.
9.	Alzughbi et al. [23], Saudi Arabia	Cross-sectional study	This study aimed to assess the prevalence of Diabetes Mellitus (DM)-related distress and depression and their associated factors in Saudi people with T2DM in Jazan, Saudi Arabia. It also aimed to assess the association between glycaemic control and DM-related distress and depression.	300 Saudi patients with T2DM randomly from primary healthcare centres in Jazan, Saudi Arabia	17-item Diabetes Distress Scale and the Patient Health Questionnaire-9	Female, patients aged < 45, physical inactivity, DM duration < 5 years, and smoking were significantly associated with DM-related distress and depression.	Biological, Social, Psychological	Cross-sectional study
10.	Sharma et al. [24], Nepal	Cross-sectional study	This study aimed to assess the depression and anxiety among patients with Type 2 Diabetes Mellitus in Chitwan.	296 Type 2 Diabetes patients admitted in the Chitwan Medical College Teaching Hospital	Patient Health Questionnaire-9 (PHQ-9)	Educational status, smoking habit, satisfaction toward current treatment, and history of diabetes in family were the factors associated with depression	Social, Biological, Psychological	(1) It is a cross-sectional study, which could not explore the causal relationship between anxiety and depression with other associated factors. (2) The study was conducted among diabetes patients who were admitted in tertiary care hospital setting, which may itself mean higher anxiety and depression. (3) It did not exclude the patients with chronic complications, which might have influenced the study findings.
11.	Alzahrani et al. [25], Saudi Arabia	Cross-sectional study	To investigate the prevalence and predictors of depression, anxiety, and stress among T2DM patients in the western region of Saudi Arabia	450 adults with T2DM	Depression, Anxiety, and Stress Scale (DASS-21) questionnaire	Presence of comorbidity, female	Biological	The cross-sectional design is inadequate to assess the direction of the relationship between depression, anxiety, and stress and T2DM.
12.	Kim et al. [26], Korea	Cross-sectional study	To evaluate the relationship between T2DM-related factors and the degree of depression based on gender in elderly patients with T2DM. We also evaluated and considered other possible factors that can affect depression, such as cognitive function, physical function, education level, and other personal factors.	155 patients with T2DM	The Geriatric Depression Scale-Korean version (SGDS-K)	Poorer glycemic control and a longer duration of DM in elderly male patients with T2DM.	Biological	(1) The study was cross sectional; (2) The sample size was relatively small, and the participants were enrolled from several general hospitals, raising concern about the generalizability of the results. (3) They relied on the SGDS-K without an accompanying diagnostic interview by a professional
13.	Victoria and Dampil [27], Philippines	Cross-sectional study	The objective of the study was to determine the prevalence of depression among adult Filipino patients with Type 2 Diabetes Mellitus and investigate the different clinical factors associated with it.	476 patients aged above 18 years old diagnosed with Type 2 Diabetes Mellitus	Patient Health Questionnaire 9 (PHQ-9)	Post-graduate degree, presence of retinopathy, and higher MMA Score (lower adherence)	Biological, Social	(1) Samples were recruited from 1 hospital only, (2) complications of diabetes were noted per chart review and as reported by the subjects only, (3) this study is cross-sectional, and a causal relationship between diabetes and depression could not be established.
14.	Radzi et al. [28], Malaysia	Cross-sectional study	To determine the prevalence of depression and its associated factors among elderly with Type 2 Diabetes Mellitus in Kedah	511 patients	The Malay version of Geriatric Depression Scale (M-GDS-14)	Living arrangements, diabetic complication and HbA1c	Environmental, Biological	This study only explored the sociodemographic, living arrangements, and diabetes status, i.e., duration of diabetes, HbA1c, co-morbidities, and diabetic complications in general.
15.	Fung et al. [29], China	Cohort	To examine the associations of depression using Geatric Depression Scale with control of cardiometabolic risk factors and health status in elderly patients with T2DM	325 participants	The Traditional Chinese version of the 15-item Geatric Depression Scale (GDS)	Presence of comorbidities	Biological	-
16.	Yoong et al. [30], Singapore	Cross-sectional study	To compare anxiety and/or depressive symptoms between patients with end-stage renal disease with and without comorbid diabetes, and identify factors associated with symptoms of distress in this population	526 patients on haemodialysis (68.8% with diabetes)	Hospital Anxiety and Depression Scale (HADS)	Single/unpartnered, Chinese, lower albumin level (poor nutritional status)	Social, Biological, Nutritional	(1) Screening for eligibility was based on medical history with no formal cognitive diagnostic evaluation, (2) case was based on self-reported data
17.	Atif et al. [31], Pakistan	Cross-sectional study	To assess the extent of depression and mild cognitive impairment (MCI) and their possible determinants among the elderly with Type 2 Diabetes Mellitus in Pakistan	490 elderly patients with Type 2 Diabetes Mellitus	Geriatric Depression Scale (GDS-15)	High HbA1C and mild cognitive Impairment were significant predictors of depression	Biological and Psychological	Limited to elderly patients only
18.	Albasheer et al. [32], Saudi Arabia	Cross-sectional study	To determine the prevalence of depression and related risk factors among Type 2 Diabetes Mellitus patients (T2DM) in the Jazan area, Saudi Arabia	400 participants	Patient Health Questionnaire (PHQ-9)	Presence of diabetic foot, cardio-vascular diseases, eye complication, and erectile dysfunction	Biological	Other variables, such as family history of depression or other psychiatric illness, previous diagnosis of depression, and drug history, were not included in this study.
19.	Kayar et al. [33], Turkey	Cross-sectional study	To investigate the relationship between depression and demographic and anthropometric characteristics, poor glycaemic control, and duration of diabetes in patients with Type 2 DM. The second question deals with the relationship between depression and individual lifestyle factors, and the third question investigates the relationship between depression and health complications.	154 patients with type 2 DM	The SCID-I scales (The Structured Clinical Interview For DSMIV, Axis I disorder) was administered to all patients by a psychiatrist to detect the presence of depression	Gender, duration of disease, hypertension, and poor glycaemic control	Biological, Social, Psychological	Cross-sectional study

**Table 3 ijerph-19-04888-t003:** Factors associated with depression in the included studies.

Factors Associated with Depression		Studies
Biological	Age	Al-Ozairi et al. [19]
	Gender (Female)	Al-Ozairi et al. [19], Alzahrani et al. [25], Abate and Gedamu [6], Majumdar et al. [17], Pal et al. [16]
	Long duration of diabetes	Kim et al. [26]
	Body mass index	Al-Ozairi et al. [19], Kant et al. [18]
	HbA1c	Al-Ozairi et al. [19], Atif et al. [31], Kant et al. [18], Kayar et al. [33], Kim et al. [26], Radzi et al. [28]
	Presence of comorbidities	Fung et al. [29], Kayar et al. [33], Radzi et al. [28], Victoria and Dampil [27]
	Diabetic complications	Albasheer et al. [32], Kayar et al. [33], Pal et al. [16], Victoria and Dampil [27]
	Sexual dysfunction	Abate and Gedamu [6]
Psychological	Fear of diabetic complications, anxiety	Abate and Gedamu [6], Woon et al. [22]
	Mild impairment cognitive	Atif et al. [31]
	Low satisfaction withcurrent treatment	Sharma et al. [24]
	Physical inactivity	Alzughbi et al. [23], Gebre et al. [20]
Social	Consumption of alcohol	Gebre et al. (2020)
	Poor social support	Abate and Gedamu [6], Gebre et al. [20]
	Low educational status	Gupta et al. [21], Sharma et al. [24], Victoria and Dampil [27]
	Smoking habit	Alzughbi et al. [23], Sharma et al. [24]
	Marital status, Single/Unpartnered	Al-Ozairi et al. [19], Yoong et al. [30]
	Ethnicity (Chinese)	Yoong et al. [30]
	Living arrangement	Radzi et al. [28]
	Low socioeconomic status	Majumdar et al. [17]
Nutritional	Lower albumin level (poor nutritional status)	Yoong et al. [30]

## Data Availability

Not applicable.

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
