# Peer review of "Biopsychosocial and Nutritional Factors of Depression among Type 2 Diabetes Mellitus Patients: A Systematic Review"

_ijerph, 2022, doi:10.3390/ijerph19084888_

Round 1

Reviewer 1 Report

The paper submitted for the review entitled ‘Biopsychosocial and nutritional factors of depression in patients with type 2 diabetes; A systematic review’ is a meta-analysis that attempts to explain the mechanisms of depression development in patients with type 2 diabetes. The occurrence of depression in patients with type 2 diabetes has been reported in numerous publications for several decades. Studies show that this psychological disorder is more common in diabetics than in non-diabetics. There are many factors that determine the development of depression associated with type 2 diabetes. This study, based on a review of 19 publications, is needed because it describes in a fairly comprehensive manner the links between depression in patients with type 2 diabetes and biological, psychological, social and nutritional factors. In my opinion, this study has some important implications useful in clinical practice.
The review of the world literature on the subject was based on the PRISMA protocol and AXIS tool, which are appropriate, recognized and used in meta-analytical studies tools.
According to my assessment, the weaknesses of the work include:

Major issues

1.    A weakness of the paper is the limitation of the literature review from the last few years (2017-2021). Findings including the prevalence of depression in patients with type 2 diabetes were presented as early as the 1980s and the associations of depression in such patients were described before 2017; example publications: Habtewold TD et al. Comorbidity of depression and diabetes: an application of biopsychosocial model. Int J Ment Health Syst. 2016;10:74., Ell K et al. Demographic, clinical and psychosocial factors identify a high-risk group for depression screening among predominantly Hispanic patients with Type 2 diabetes in safety net care. Gen Hosp Psychiatry. 2015;37(5):414-9. 
2.    The main research question "What are the factors associated with depression among T2DM patients?" is too vague as it does not specify that it is about factors that constitute the risk of developing depression in patients with type 2 diabetes. 
3.    A shortcoming of the paper is the inclusion in the selection of publications only articles available in open-access. 

Minor issues

1.    There is a noticeable inconsistency between the statement of purpose and the main conclusion. The conclusion "This systematic review identified multiple factors that contributed to depression among T2DM patients from multiple countries. The findings revealed that biopsychosocial and nutritional factors had a greater impact on depression among T2DM patients" should correspond with the stated aim. In this context, the content of the aim of the study should be corrected.
2.    Among biological factors, the authors list demographic indicators (age, gender, marital status). It is worth considering separating them.
3.    Lifestyle components have been included as psychological factors. Please justify this.
4.    Mild impairment of cognitive function, are among biological factors, but it is a psychological disorder.

Other remarks

The Eagle citation refers to the item 8 and should be 9 according to the cited literature (line 46). The citations of Taukeni (8th position in the References) and Engel (9th position in the References) should be explained.
In lines 107-109, the text is written as if it were the title of Table 2. This should be corrected.
In Table 3, in the "Determinants" column in item 18 (Yoong et al), it seems that the term "Nutritional" should be included. According to the cited Yoong et al. the relationship between depression and nutrition is placed in Table 4. The Habtewold et al. citation needs to be annotated with 4 instead of 2016 (lines 151 and 190)
References need to be corrected according to the journal rules. Items 9, 14, 16, 18, 19-22, 26, 28-30, 32, 40 52, 53, 58, 63, 68 are missing the journal name. Items 13, 15, 18-20, 23, 29, 40, 52, 67 are missing chapter and page numbers. Numerous punctuation corrections are required.

Author Response

Dear Sir, 

Thank you for your comments. Kindly please see the attachment. 

Reviewer 2 Report

This article reviews the biopsychosocial and nutritional aspects related to depression in people with type 2 diabetes. An interesting premise for including various areas important for the development or maintenance of depression in people with type 2 diabetes.

However, there are several aspects to improve or change in the article:

- Introduction: Lines 33 and 34. This sentence is not understood.

As for the link between the sentences on lines 41, 42 and 43, it is not correct. It should be redrafted.

- Material and Methods: Reference and detailed explanation of PICO concept.

 The review is done between 2017 and 2021. Why has this small range been chosen? Justify.

Table 2. Replace this table with explanation in the text. It is not necessary to have this information in the article (it can be added as an Appendix if you want to include it).

What does it mean to obtain a score of 16 (range: 15-17) for the average quality of articles (quality, high, medium…)? Explain in the text.

Who did the process of searching and selecting articles? As explained in section 2.7. Data Abstraction and Analysis is understood to have been done by the three authors and put together. Explain it better in the text.

- Results.

Table 3. The headings Factors associated and Determinants are not well understood to what they refer. Explain epigraphs collected in text.

Questionnaires that are not about depression should not be included in the Depression Measurement section. In addition, article number 14 does not contain any questionnaire that assesses depression.

In Table 4 (it comes from Table 3 and is included in the text on several occasions) there is talk of Fears, what type of Fears are the authors referring to? Specify.

- Discussion: line 151. Number the reference. Review and do the same in the rest of the article's references (including tables).

Line 167. Very short phrase. Make better the link between the previous and subsequent sentences.

Line 183. Confuses the union of ideas between different factors. Put in order and separate by paragraphs.

Line 288-289. "This review included only open-access articles." why? This should be explained in Material and Methods.

- References. There are many incompletes (for example: 14, 18, 28-30, 32, 40, 58…). Review all and complete following the rules of the magazine.

Why does it say “Available from” on some and not on others? See journal standards and unify references.

Author Response

Dear Sir,

Thank you for your comments. Kindly please see the attachment. Thank you.

Round 2

Reviewer 1 Report

The answers provided and the corrections made are to my satisfaction. However, in Table 2 in the column "Author (Year) Country" should be removed (Year) and in items 7, 10 and 14 should be written authors.

Author Response

Thank you for the comments & suggestions. 

  1. In Table 2 in the column "Author (Year) Country" should be removed (Year):
    • '(Year)' has been removed as per your suggestion.
  2. Items 7, 10 and 14 should be written authors:
    • All items in Table 2 have been rechecked, and the authors' names have been edited according to the citation format, as follows:
      • Item 7: "Abate et al." changed to "Abate and Gedamu"(only 2 authors).  
      • Item 10: ''Sharma et al.'' (several writing authors).
      • Item 12: "Kim et al." changed to "Kim" (only 1 author).
      • Item 13: "Victoria et al." changed to "Victoria and Dampil" (only 2 authors).
      • Item 14: "Radzi et al." (several writing authors).

Thank you again.

Reviewer 2 Report

Table 1. Author, year and country: review this part. 

Author Response

Thank you for the comments & suggestions. 

  1. Table 1. Author, year and country: review this part:
    • Perhaps you are referring to Table 2.
    • All items in Table 2 have been rechecked. The title of the column has been changed from "Author (Year), Country" to "Authors, Country", as per suggested by another reviewer.
    • The authors' names have been edited according to the citation format, as follows:
      • Item 7: "Abate et al." changed to "Abate and Gedamu" (only 2 authors).
      • Item 10: ''Sharma et al.'' (several writing authors).
      • Item 12: "Kim et al." changed to "Kim" (only 1 author).
      • Item 13: "Victoria et al." changed to "Victoria and Dampil" (only 2 authors).
      • Item 14: "Radzi et al." (several writing authors).

Thank you again.